# Deep Signature Transforms

**Patric Bonnier**[1,*]    **Patrick Kidger**[1,2,*]    **Imanol Perez Arribas**[1,2,*]

**Cristopher Salvi**[1,2,*]    **Terry Lyons**[1,2]

[1] Mathematical Institute, University of Oxford
[2] The Alan Turing Institute, British Library
{bonnier, kidger, perez, salvi, tlyons}@maths.ox.ac.uk

## Abstract

The signature is an infinite graded sequence of statistics known to characterise a stream of data up to a negligible equivalence class. It is a transform which has previously been treated as a fixed feature transformation, on top of which a model may be built. We propose a novel approach which combines the advantages of the signature transform with modern deep learning frameworks. By learning an augmentation of the stream prior to the signature transform, the terms of the signature may be selected in a data-dependent way. More generally, we describe how the signature transform may be used as a layer anywhere within a neural network. In this context it may be interpreted as a pooling operation. We present the results of empirical experiments to back up the theoretical justification. Code available at `github.com/patrick-kidger/Deep-Signature-Transforms`.

## 1 Introduction

### 1.1 What is the signature transform?

When data is ordered sequentially then it comes with a natural path-like structure: the data may be thought of as a discretisation of a path $X\colon [0, 1] \to V$, where $V$ is some Banach space. In practice we shall always take $V = \mathbb{R}^d$ for some $d \in \mathbb{N}$. For example the changing air pressure at a particular location may be thought of as a path in $\mathbb{R}$; the motion of a pen on paper may be thought of as a path in $\mathbb{R}^2$; the changes within financial markets may be thought of as a path in $\mathbb{R}^d$, with $d$ potentially very large.

Given a path, we may define its *signature*, which is a collection of statistics of the path. The map from a path to its signature is called the *signature transform*.

**Definition 1.1.** Let $\mathbf{x} = (x_1, \ldots, x_n)$, where $x_i \in \mathbb{R}^d$. Let $f = (f_1, \ldots, f_d)\colon [0, 1] \to \mathbb{R}^d$ be continuous, such that $f(\frac{i-1}{n-1}) = x_i$, and linear on the intervals in between. Then the signature of $\mathbf{x}$ is defined as the collection of iterated integrals[2]

$$\mathrm{Sig}(\mathbf{x}) = \left( \left( \underset{0 < t_1 < \cdots < t_k < 1}{\int \cdots \int} \prod_{j=1}^{k} \frac{\mathrm{d}f_{i_j}}{\mathrm{d}t}(t_j)\mathrm{d}t_1 \cdots \mathrm{d}t_k \right)_{1 \le i_1, \ldots, i_k \le d} \right)_{k \ge 0}.$$

---

[*]Equal contribution.

[2]For clarity here we have used more widely-understood notation. The definition of the signature transform is usually written in an equivalent but alternate manner using the notation of stochastic calculus; see Definition A.1 in Appendix A.

We shall often use the term *signature* to refer to both a path's signature and the signature transform. Other texts sometimes use the term *path signature* in a similar manner.

We refer the reader to [1] for a primer on the use of the signature in machine learning. A brief overview of its key properties may be found in Appendix A, along with associated references.

In short, the signature of a path determines the path essentially uniquely, and does so in an efficient, computable way. Furthermore, the signature is rich enough that every continuous function of the path may be approximated arbitrarily well by a linear function of its signature; it may be thought of as a 'universal nonlinearity'. Taken together these properties make the signature an attractive tool for machine learning. The most simple way to use the signature is as feature transformation, as it may often be simpler to learn a function of the signature than of the original path.

Originally introduced and studied by Chen in [2, 3, 4], the signature has seen use in finance [5, 6, 7, 8, 9], rough path theory [10, 11] and machine learning [12, 13, 14, 15, 16, 17, 18, 19, 20].

## 1.2 Comparison to the Fourier transform

The signature transform is most closely analogous to the Fourier transform.

The fundamental difference between the signature transform and classical signal transforms such as Fourier transforms and wavelets is that the latter are used to model a curve as a linear combination in a functional basis. The signature does not try to model or parameterise the curve itself, but instead provides a basis for functions on the space of curves.

For example, regularly seeing the sequence: phone call, trade, price movement in the stream of office data monitoring a trader might be an indication of insider trading. Such occurrences are straightforward to detect by via a linear regression composed with the signature transform. Modelling this signal using Fourier series or wavelets would be much more expensive: linearity of these transforms imply that each channel must be resolved accurately enough to see the order of events.

From a signal processing perspective, the signature can be thought of as a filter which is invariant to resampling of the input signal. (See Proposition A.7 in Appendix A).

## 1.3 Use of the signature transform in machine learning

The signature is an infinite sequence, so in practice some finite collection of terms must be selected. Since the magnitude of the terms exhibit factorial decay, see Proposition A.5 in Appendix A, it is usual [21] to simply choose the first $N$ terms of this sequence, which will typically be the largest terms . These first $N$ terms are called the *signature of depth $N$* or the *truncated signature of depth $N$*, and the corresponding transform is denoted $\mathrm{Sig}^N$. But if the function to be learned depended nontrivially on the higher degree terms, then crucial information has nonetheless been lost.

This may be remedied. Apply a pointwise augmentation to the original stream of data before taking the signature. Then the first $N$ terms of the signature may better encode the necessary information [19, 20]. Explicitly, let $\Phi \colon \mathbb{R}^d \to \mathbb{R}^e$ be fixed; one could ensure that information is not lost by taking $\Phi(x) = (x, \varphi(x))$ for some $\varphi$. Then rather than taking the signature of $\mathbf{x} = (x_1, \ldots, x_n)$, where $x_i \in \mathbb{R}^d$, instead take the signature of $\Phi(\mathbf{x}) = (\Phi(x_1), \ldots, \Phi(x_n))$. In this way one may capture higher order information from the stream in the lower degree terms of the signature.

## 1.4 Our work

But how should this augmentation $\Phi$ be chosen? Previous work has fixed it arbitrarily, or experimented with several options before choosing one [19, 20]. Observe that in each case the map $\mathbf{x} \mapsto \mathrm{Sig}^N(\Phi(\mathbf{x}))$ is still ultimately just a feature transformation on top of which a model is built. Our more general approach is to allow the selection of $\Phi$ to be data-dependent, by having it be learned; in particular it may be a neural network. Furthermore there is no reason it should necessarily operate pointwise, nor (since it is now learned) need it be of the form $(x, \varphi(x))$. In this way we may enjoy the benefits of using signatures while avoiding their main limitation.

But this means that the signature transform is essentially operating as a layer within a neural network. It consumes a tensor of shape $(b, d, n)$ – corresponding to a batch of size $b$ of paths in $\mathbb{R}^d$ that have

been sampled $n$ times – and returns a tensor of shape $(b, (d^{N+1}-1)/(d-1))$, where $N$ is the number of terms used in the truncated signature.[3] The signature is being used as a pooling operation.

There is no reason to stop here. If the signature layer works well once then it is natural to seek to use it again. The obvious problem is that the signature transform consumes a stream of data and returns statistics which have no obvious stream-like qualities. The solution is to lift the input stream to a *stream of streams*; for example, the stream of data $(x_1, \ldots, x_n)$ may be lifted to the 'expanding windows' of $(\mathbf{x}_2, \ldots, \mathbf{x}_n)$, where $\mathbf{x}_i = (x_1, \ldots, x_i)$. Now apply the signature to each stream to obtain a stream of signatures $(\mathrm{Sig}^N(\mathbf{x}_2), \ldots, \mathrm{Sig}^N(\mathbf{x}_n))$, which is essentially a stream in Euclidean space. And now this new stream may be augmented via a neural network and the process repeated again, as many times as we wish.

In this way the signature transform has been elevated from a one-time feature transformation to a first-class layer within a neural network. Thus we may reap the benefits of both the signature transform, with its strong corpus of mathematical theory, and the benefits of neural networks, with their great empirical success.

Naturally all of this implies the need for an efficient implementation of the signature transform. Such concerns have motivated the creation of the spin-off Signatory project [22].

The remainder of the paper is laid out as follows. In Section 2 we briefly discuss some related work, in Section 3 we detail the specifics of embedding the signature as a layer within a neural network. Sections 4 covers experiments; we demonstrate positive results for generative, supervised, and reinforcement learning problems. Section 5 is the conclusion. Appendix A provides an exposition of the theoretical properties of the signature, and Appendix B specifies implementation details.

## 2   Related Work

The signature transform is roughly analogous to the use of wavelets or Fourier transforms, and there are also related models based around these, for example [23, 24, 25, 26]. We do not know of a detailed comparison between the use of these various transformations in the context of machine learning.

Some related work using signatures has already been discussed in the previous section. We expand on their proposed models here.

**Definition 2.1.** Given a set $V$, the space of streams of data in $V$ is defined as

$$\mathcal{S}(V) = \{\mathbf{x} = (x_1, \ldots, x_n) : x_i \in V, n \in \mathbb{N}\}.$$

Given $\mathbf{x} = (x_1, \ldots, x_n) \in \mathcal{S}(V)$, the integer $n$ is called the length of $\mathbf{x}$.

Two simple models utilising the signature layer are shown in Figure 1.

In principle the universal nonlinearity property of signatures (see Proposition A.6 in Appendix A) guarantees that the model shown in Figure 1a, is rich enough to learn any continuous function. (With the neural network taken to be a single linear layer and the input stream assumed to already be time-augmented.) In practice, of course, the signature must be truncated. Furthermore, it is not clear how to appropriately choose the truncation hyperparameter $N$. Thus a more practical approach is to remove the restriction that the neural network must be linear, and learn a nonlinear function instead. This approach has been applied successfully in various tasks [5, 12, 13, 14, 15, 16, 17, 18].

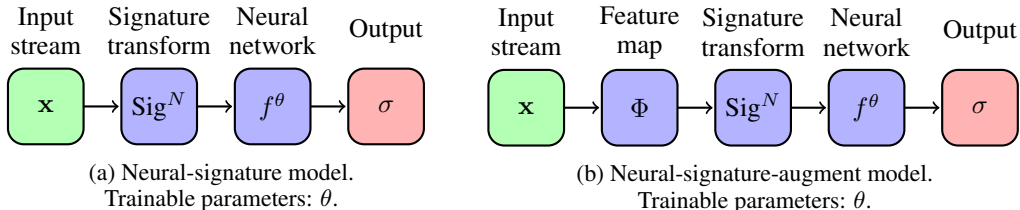

(a) Neural-signature model.
Trainable parameters: $\theta$.

(b) Neural-signature-augment model.
Trainable parameters: $\theta$.

Figure 1: Two simple architectures with a signature layer.

An alternate model is shown in Figure 1b. Following [19, 20], a pointwise transformation could be applied to the stream before taking the signature transform. That is, applying the feature map $\Phi\colon \mathbb{R}^d \to \mathbb{R}^e$ to the $d$-dimensional stream of data $(x_1, \ldots, x_n) \in \mathcal{S}(\mathbb{R}^d)$ yields $(\Phi(x_1), \ldots, \Phi(x_n)) \in \mathcal{S}(\mathbb{R}^e)$; the signature of $\Phi(\mathbf{x})$ may then potentially capture properties of the stream of data that will yield more effective models.

## 3 The signature transform as a layer in a neural network

However, there is not always a clear candidate for the feature map $\Phi$ and a good choice is likely to be data-dependent. Thus we propose to make $\Phi$ learnable by taking $\Phi = \Phi^\theta$ to be a neural network with trainable parameters $\theta$. In this case, we again obtain the neural network shown in Figure 1b, except that $\Phi$ is now also learnable.

The signature has now become a layer within a neural network. It consumes a tensor of shape $(b, d, n)$ – corresponding to a batch of size $b$ of paths in $\mathbb{R}^d$ that have been sampled $n$ times – and returns a tensor of shape $(b, (d^{N+1} - 1)/(d - 1))$, where $N$ is the number of terms used in the truncated signature.

Despite being formed of integrals, the signature is in fact straightforward and efficient to compute exactly, see Section A.3 in Appendix A. More than that, the computation may in fact be described in terms of standard tensor operations. As such it may be backpropagated through without difficulty.

### 3.1 Stream-preserving neural networks

Let $\mathbf{x} = (x_1, \ldots, x_n) \in \mathcal{S}(\mathbb{R}^d)$. Whatever the choice of $\Phi^\theta$, it must preserve the stream-like nature of the data if we are to take a signature afterwards. The simplest way of doing this is to have $\Phi^\theta$ map $\mathbb{R}^d \to \mathbb{R}^e$, so that it operates pointwise. This defines $\Phi(\mathbf{x})$ by

$$\Phi(\mathbf{x}) = (\Phi^\theta(x_1), \ldots, \Phi^\theta(x_n)) \in \mathcal{S}(\mathbb{R}^e). \tag{1}$$

Another way to preserve the stream-like nature is to sweep a one dimensional convolution along the stream; more generally one could sweep a whole feedforward network along the stream. For some $m \in \mathbb{N}$ and $\Phi^\theta\colon \mathbb{R}^{d \times m} \to \mathbb{R}^e$ this defines $\Phi(\mathbf{x})$ by

$$\Phi(\mathbf{x}) = (\Phi^\theta(x_1, \ldots, x_m), \ldots, \Phi^\theta(x_{n-m+1}, \ldots, x_n)) \in \mathcal{S}(\mathbb{R}^e). \tag{2}$$

More generally still the network could be recurrent, by having memory. Let $\Phi_0 = 0$, fix $m \in \mathbb{N}$, and define $\Phi_k = \Phi^\theta(x_k, \ldots, x_{k+m}; \Phi_{k-1})$ for $k = 1, \ldots, n - m + 1$. Then define $\Phi(\mathbf{x})$ by

$$\Phi(\mathbf{x}) = (\Phi_1, \ldots, \Phi_{n-m+1}) \in \mathcal{S}(\mathbb{R}^e). \tag{3}$$

### 3.2 Stream-like data

It is worth taking a moment to think what is really meant by 'stream-like nature'. The signature transform is defined on paths; it is applied to a stream of data in $\mathcal{S}(\mathbb{R}^d)$ by first interpolating the data into a path and then taking the signature.

The data is treated as a discretisation or set of observations of some underlying path. Note that there is nothing wrong with the path itself having a discrete structure to it; for example a sentence.

In principle one could reshape a tensor of shape $(b, nd)$ with no stream-like nature into one of shape $(b, d, n)$, and then take the signature. However it is not clear what this means mathematically. There is no underlying path. The signature is at this point an essentially arbitrary transformation, without the mathematical guarantees normally associated with it.

### 3.3 Stream-preserving signatures, using lifts

We would like to apply the signature layer multiple times. However applying the signature transform consumes the stream-like nature of the data, which prevents this. The solution is to construct a stream of signatures in the following way: given a stream $\mathbf{x} = (x_1, \ldots, x_n) \in \mathcal{S}(\mathbb{R}^d)$, let $\mathbf{x}_k = (x_1, \ldots, x_k)$ for $k = 2, \ldots, n$, and apply the signature to each $\mathbf{x}_k$ to obtain the stream

$$(\mathrm{Sig}^N(\mathbf{x}_2), \ldots, \mathrm{Sig}^N(\mathbf{x}_n)) \in \mathcal{S}(\mathbb{R}^{(d^{N+1}-1)/(d-1)}). \tag{4}$$

The shortest stream it is meaningful to take the signature of is of length two, which is why there is no corresponding $\mathrm{Sig}^N(\mathbf{x}_1)$ term.

In this way the stream-like nature of the data is preserved through the signature transform.

This notion may be generalised: let

$$\ell = (\ell^1, \ell^2, \ldots, \ell^v) \colon \mathcal{S}(\mathbb{R}^d) \to \mathcal{S}(\mathcal{S}(\mathbb{R}^e)),$$

which we refer to as a *lift* into the space of streams of streams (and $v$ will likely depend on the length of the input to $\ell$). Then apply the signature stream-wise to define $\mathrm{Sig}^N(\ell(\mathbf{x}))$ by

$$\mathrm{Sig}^N(\ell(\mathbf{x})) = \left(\mathrm{Sig}^N(\ell^1(\mathbf{x})), \ldots, \mathrm{Sig}^N(\ell^v(\mathbf{x}))\right) \in \mathcal{S}(\mathbb{R}^{(e^{N+1}-1)/(e-1)}). \tag{5}$$

In the example of equation (4), $\ell$ is given by

$$\ell(\mathbf{x}) = (\mathbf{x}_2, \ldots, \mathbf{x}_n). \tag{6}$$

Other plausible choices for $\ell$ are to cut up $\mathbf{x}$ into multiple pieces, for example

$$\ell(\mathbf{x}) = ((x_1, x_2), (x_3, x_4), \ldots, (x_{2\lfloor n/2\rfloor-1}, x_{2\lfloor n/2\rfloor})), \tag{7}$$

or to take a sliding window

$$\ell(\mathbf{x}) = ((x_1, x_2, x_3), (x_2, x_3, x_4), \ldots, (x_{n-2}, x_{n-1}, x_n)). \tag{8}$$

### 3.4 Multiple signature layers

By inserting lifts, the signature transform may be composed as many times as desired. That is, suppose we wish to learn a map from $\mathcal{S}(\mathbb{R}^d)$ to $\mathcal{X}$, where $\mathcal{X}$ is some set. (Which may be finite for a classification problem or infinite for a regression problem.) Let $c_i, d_i, e_i, N_i \in \mathbb{N}$ be such that $d_1 = d$ and $d_{i+1} = (c_i^{N_i+1} - 1)/(c_i - 1)$, for $i = 1, \ldots, k$.

Let

$$\Phi_i^{\theta_i} \colon \mathbb{R}^{d_i \times m_i} \to \mathbb{R}^{e_i}, \qquad \ell_i \colon \mathcal{S}(\mathbb{R}^{e_i}) \to \mathcal{S}(\mathcal{S}(\mathbb{R}^{c_i})), \qquad f^{\theta_{k+1}} \colon \mathcal{S}(\mathbb{R}^{(c_k^{N_k+1}-1)/(c_k-1)}) \to \mathcal{X},$$

where $\Phi_i^{\theta_i}$ and $\ell_i$ are defined in the manner of equations (1)–(3) and (6)–(8), and $\theta_1, \ldots, \theta_{k+1}$ are some trainable parameters. Then defining compositions in the manner of equations (1)–(5), let

$$\sigma = \left(f^{\theta_{k+1}} \circ \mathrm{Sig}^{N_k} \circ \ell_k \circ \Phi_k^{\theta_k} \circ \cdots \circ \Phi_2^{\theta_2} \circ \mathrm{Sig}^{N_1} \circ \ell_1 \circ \Phi_1^{\theta_1}\right)(\mathbf{x}).$$

This defines the *deep signature model*, summarised in Figure 2.

An important special case is when $V = \mathcal{S}(\mathbb{R}^e)$, so that the final network $f^{\theta_{k+1}}$ is stream-preserving. Then the overall model $\mathbf{x} \mapsto \sigma$ is also stream-preserving. See for example Section 4.1.

Note that in principle it is acceptable to take the trivial lift to a sequence of a single element,

$$\ell(\mathbf{x}) = (\mathbf{x}). \tag{9}$$

Taking the signature of this will then essentially remove the stream-like nature, however, so it is suitable only for the final lift of a deep signature model. We observe in particular that this is what is done in the models described in Figure 1, which we identify as special cases of the deep signature model, lacking also any learned transformation before the signature.

It is easy to see that the deep signature model exhibits the universal approximation property. This fact follows from the universal approximation theorem for neural networks [27] and from the universal nonlinearity property of signatures (see Proposition A.6 in Appendix A).

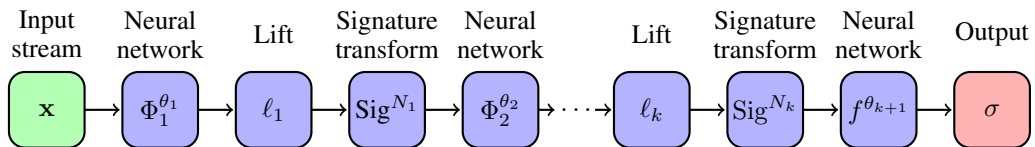

Figure 2: Deep signature model. Trainable parameters: $\theta_1, \ldots, \theta_{k+1}$.

## 3.5 Implementation

When using the signature transform as a feature transformation, then it suffices to just pre-process and save the entire dataset before training. However when the signature transform is placed within a neural network then the signature transform must be evaluated and backpropagated through for each step of training; this is much more computationally intensive. This has motivated the creation of the separate spin-off Signatory project [22], to efficiently perform and backpropagate through the signature transform.

## 3.6 Inverting the truncated signature

How well does a truncated signature encode the original stream of data? A simple experiment is to attempt to recover the original stream of data given its truncated signature. We remark that finding a mathematical description of this inversion is a challenging task [28, 29, 30].

Fix a stream of data $\mathbf{x} = (x_1, \ldots, x_n) \in \mathcal{S}(\mathbb{R}^d)$. Assume that the truncated signature $\mathrm{Sig}^N(\mathbf{x})$ and the number of steps $n \in \mathbb{N}$ are known. Now apply gradient descent to minimise

$$L(\mathbf{y}; \mathbf{x}) = \left\| \mathrm{Sig}^N(\mathbf{y}) - \mathrm{Sig}^N(\mathbf{x}) \right\|_2^2 \quad \text{for } \mathbf{y} = (y_1, \ldots, y_n) \in \mathcal{S}(\mathbb{R}^d).$$

Figure 3 shows four handwritten digits from the PenDigits dataset [31]. The solid blue path is the original path $\mathbf{x}$, whilst the dashed orange path is the reconstructed path $\mathbf{y}$ minimising $L(\mathbf{y}; \mathbf{x})$. Truncated signatures of order $N = 12$ were used for this task. We see that the truncated signatures have managed to encode the input paths $\mathbf{x}$ almost perfectly.

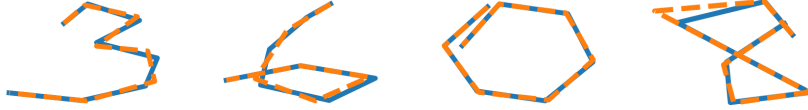

Figure 3: Original path (blue) and path reconstructed from its signature (dashed orange) for four handwritten digits in the PenDigits dataset [31].

# 4 Numerical experiments

## 4.1 A generative model for a stochastic process

Generative models are typically trained to learn to transform random noise to a target distribution. One common approach are Generative Adversarial Networks [32]. An alternative approach is to define a distance on the space of distributions by embedding them into a Reproducing Kernel Hilbert Space. The discriminator is then a fixed two-sample test based on a kernel maximum mean discrepancy. This is known as a Generative Moment Matching Network [33, 34, 35].

With this framework we propose a deep signature model to generate sequential data. The discriminator is as in [19, 20]. The natural choice for random noise is Brownian motion $B_t$.

Define the kernel $k \colon \mathcal{S}(\mathbb{R}^d) \times \mathcal{S}(\mathbb{R}^d) \to \mathbb{R}$ by

$$k(\mathbf{x}, \mathbf{y}) = \left( \mathrm{Sig}^N(\lambda_{\mathbf{x}} \mathbf{x}), \mathrm{Sig}^N(\lambda_{\mathbf{y}} \mathbf{y}) \right),$$

where $\lambda_{\mathbf{x}} \in \mathbb{R}$ is a certain normalising constant which guarantees that $k$ is the kernel of a Reproducing Kernel Hilbert Space, and $(\,\cdot\,, \cdot\,)$ denotes the dot product.

Given $n$ samples $\{\mathbf{x}^{(i)}\}_{i=1}^n \subseteq \mathcal{S}(\mathbb{R}^d)$ from the generator and $m$ samples $\{\mathbf{y}^{(i)}\}_{i=1}^m \subseteq \mathcal{S}(\mathbb{R}^d)$ from the target distribution, define the loss $T$ by

$$T\left(\{\mathbf{x}^{(i)}\}_{i=1}^n, \{\mathbf{y}^{(i)}\}_{i=1}^m\right) = \frac{1}{n^2} \sum_{i,j} k(\mathbf{x}^{(i)}, \mathbf{x}^{(j)}) - \frac{2}{nm} \sum_{i,j} k(\mathbf{x}^{(i)}, \mathbf{y}^{(j)}) + \frac{1}{m^2} \sum_{i,j} k(\mathbf{y}^{(i)}, \mathbf{y}^{(j)}).$$

Let the input to the network be time-augmented Brownian motion

$$\mathbf{B} = ((t_1, B_{t_1}), \ldots, (t_n, B_{t_n})) \in \mathcal{S}(\mathbb{R}^2).$$

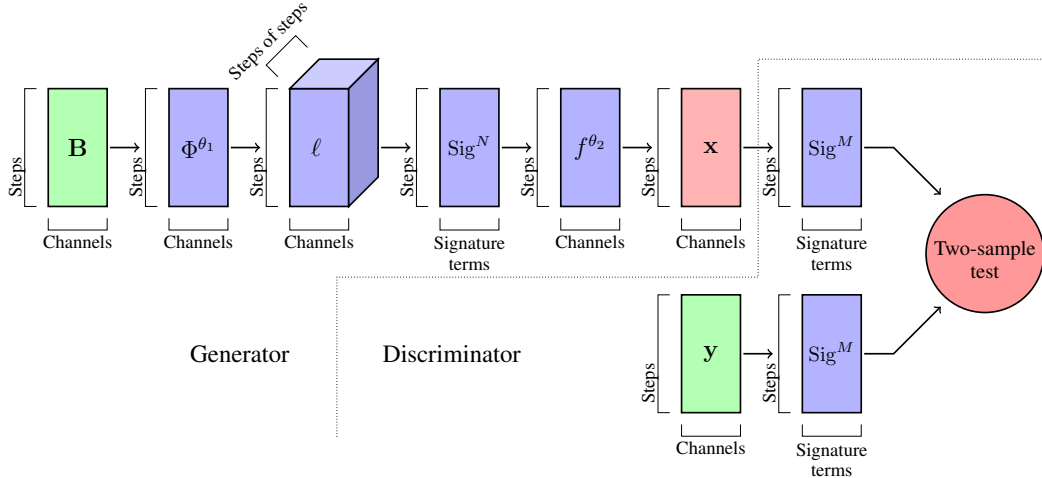

Figure 4: Generative model architecture. Trainable parameters: $\theta_1, \theta_2$. There is an implicit batch dimension throughout.

Given two stream-preserving neural networks $\Phi^{\theta_1}$ and $f^{\theta_2}$, and a lift $\ell$, then the generative model is defined by

$$\mathbf{x} = (f^{\theta_2} \circ \mathrm{Sig}^N \circ \ell \circ \Phi^{\theta_1})(\mathbf{B}).$$

The overall model is shown in Figure 4. In a nice twist, both the generator and the discriminator involve the signature.

Observe how the generative part is a particular case of the deep signature model, and that furthermore the whole generator-discriminator pair is also a particular case of the deep signature model, with the trivial lift of equation (9) before the second signature layer.

We applied the proposed model to a dataset of 1024 realisations of an Ornstein–Uhlenbeck process [36]. The loss was minimised at $6.6 \times 10^{-4}$, which implies that the generated paths are statistically almost indistinguishable from the real Ornstein–Uhlenbeck process. Figure 5 shows the generated paths alongside the original ones. Further implementation details are in Appendix B.

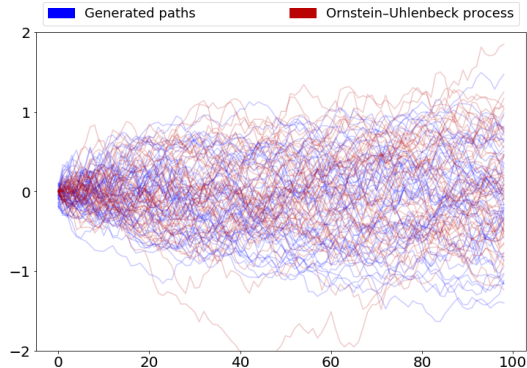

Figure 5: Generated paths alongside the original paths.

## 4.2 Supervised learning with fractional Brownian motion

Fractional Brownian motion [37] is a Gaussian process $B^H : [0, \infty) \to \mathbb{R}$ that generalises Brownian motion. It is self-similar and exhibits fractal-like behaviour. Fractional Brownian motion depends upon a parameter $H \in (0, 1)$, known as the Hurst parameter. Lower Hurst parameters result in noticeably rougher paths. The case of $H = 1/2$ corresponds to usual Brownian motion. Fractional Brownian motion has been successfully used to model phenomena in diverse fields. For example, empirical evidence from financial markets [38] suggests that log-volatility is well modelled by fractional Brownian motion with Hurst parameter $H \approx 0.1$.

Estimating the Hurst parameter of a fractional Brownian motion path is considered a nontrivial task because of the paths' non-stationarity and long range dependencies [39]. We train a variety of models to perform this estimation. That is, to learn the map $\mathbf{x}^H \mapsto H$, where

$$\mathbf{x}^H = ((t_0, B^H_{t_0}), \dots, (t_n, B^H_{t_n})) \in \mathcal{S}(\mathbb{R}^2)$$

for some realisation of $B^H$.

Table 1: Final test mean squared error (MSE) for the different models, averaged over 3 training runs, ordered from largest to smallest.

| | Test MSE | | |
| --- | --- | --- | --- |
| | Mean | Variance | # Params |
| Rescaled Range | $7.2 \times 10^{-2}$ | $3.7 \times 10^{-3}$ | N/A |
| LSTM | $4.3 \times 10^{-2}$ | $8.0 \times 10^{-3}$ | 12961 |
| Feedforward | $2.8 \times 10^{-2}$ | $3.0 \times 10^{-3}$ | 10209 |
| Neural-Sig | $1.1 \times 10^{-2}$ | $8.2 \times 10^{-4}$ | 10097 |
| GRU | $3.3 \times 10^{-3}$ | $1.3 \times 10^{-3}$ | 9729 |
| RNN | $1.7 \times 10^{-3}$ | $4.9 \times 10^{-4}$ | 10091 |
| DeepSigNet | $2.1 \times 10^{-4}$ | $8.7 \times 10^{-5}$ | 9261 |
| DeeperSigNet | $1.6 \times 10^{-4}$ | $2.1 \times 10^{-5}$ | 9686 |

The results are shown in Figure 6 and Table 1. Also shown in Table 1 are the results of the rescaled range method [40], which is a mathematically derived method rather than a learned method.

RNN, GRU and LSTM models provide baselines in the context of recurrent neural networks. The simple Neural-Sig model outlined previously in Figure 1a provides a baseline from the context of signatures.

DeepSigNet and DeeperSigNet are both deep signature models of the form given by Figure 2. DeepSigNet has a single large Neural-Lift-Signature block, whilst DeeperSigNet has three smaller ones.

We observe that traditional signature based models perform slightly worse than traditional recurrent models, but that deep signature models outperform all other models by at least an order of magnitude. Further implementation details are found in Appendix B.

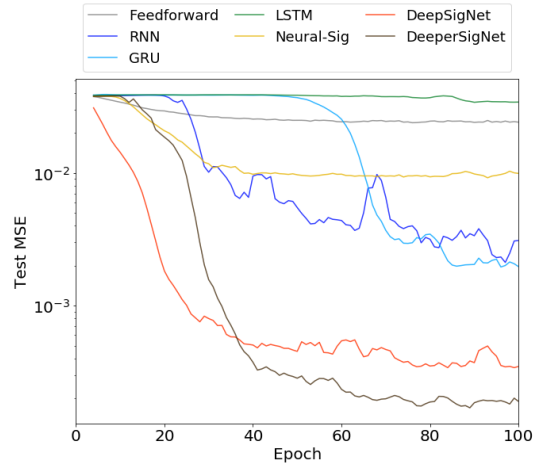

Figure 6: Performance at estimating the Hurst parameter for various models, with and without signatures, for a particular (typical) training run.

## 4.3 Non-Markovian deep reinforcement learning

Finally we show how these ideas may be extended, by demonstrating a model that adds a residual connection to the deep signature model; it may also be interpreted as using signatures as the memory of a recurrent neural network.

As an example, we apply this architecture to tackle a non-Markovian reinforcement learning problem. This means that the optimal action depends not just on the current state of the environment, but upon the history of past states, so that the agent must maintain a memory.

Let $\Phi^{\theta_1} \colon \mathbb{R}^d \to \mathbb{R}^e$ and $f^{\theta_2} \colon \mathbb{R}^{d+(e^{N+1}-1)/(e-1)} \to \{\text{actions}\}$ be functions depending on learnable parameters $\theta_1$, $\theta_2$. Given input $x_i \in \mathbb{R}^d$ at time $i$, let

$$y_i = \Phi^{\theta_1}(x_i), \qquad \sigma_i = \sigma_{i-1} \otimes \text{Sig}^N((y_{i-1}, y_i)), \qquad a_i = f^{\theta_2}(x_i, \sigma_i),$$

where $a_i$ is the action proposed by the network at time $i$, and $y_i$ and $\sigma_i$ are the memory at time $i$, and $\otimes$ denotes the tensor product as in A.13 in Appendix A.

The model is summarised in Figure 7 as a recurrent neural network with signature-based memory. Note that $y_i$ is preserved in memory only to compute the signature at the next time step, as the shortest path it is meaningful to compute the signature of is of length two.

However, note that by Proposition A.15 in Appendix A,

$$\sigma_i = \mathrm{Sig}^N(\Phi^{\theta_1}(x_1), \ldots, \Phi^{\theta_1}(x_i)) \in \mathbb{R}^{(e^{N+1}-1)/(e-1)}.$$

Furthermore the $x_i$, $y_i$, $\sigma_i$ and $a_i$ may be collected into streams

$$(x_i)_i \in \mathcal{S}(\mathbb{R}^d),$$
$$(y_i)_i \in \mathcal{S}(\mathbb{R}^e),$$
$$(\sigma_i)_i \in \mathcal{S}(\mathbb{R}^{(e^{N+1}-1)/(e-1)}),$$
$$(a_i)_i \in \mathcal{S}(\{\mathrm{actions}\}).$$

In this way we may interpret this model as a generalisation of deep signature model: it has a single Neural-Lift-Signature block, with a skip connection across the whole block. The neural component is given by the neural network $\Phi^{\theta_1}$, which is stream-preserving as it operates pointwise, in the manner of equation (1). The lift is the 'expanding window' lift given by equation (6). Finally $f^{\theta_2}$ is another neural network, which is again pointwise and thus stream-preserving.

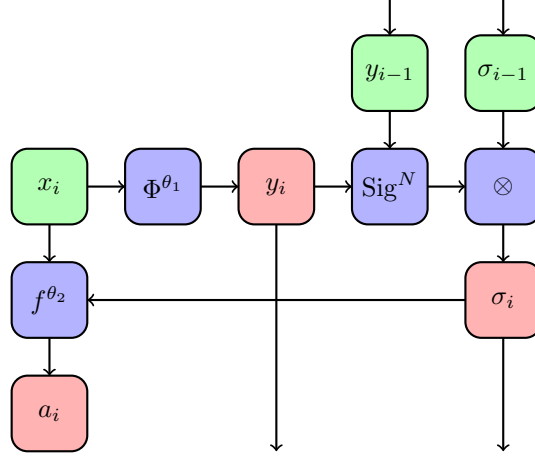

Figure 7: Agent architecture as a recurrent network. Trainable parameters: $\theta_1, \theta_2$.

This interpretation of the model is demonstrated in Figure 8.

We test this model on a non-Markovian modification to the classical Mountain Car problem [41], in which the agent receives only partial information: it is only given the car's position, and not its velocity.We find that it is capable of learning how to solve the problem within a set number of episodes, whilst a comparable RNN architecture fails to do so. The reinforcement learning technique used was Deep Q Learning [42] with the specified models performing function approximation on $Q$. Both models were chosen to have comparable numbers of parameters. Further implementation details can be found in Appendix B.

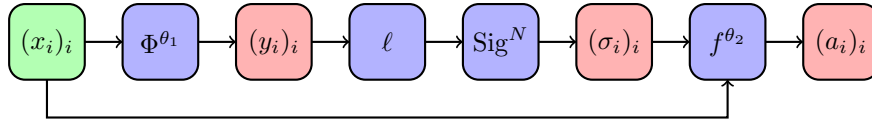

Figure 8: Agent architecture as a residual network. Trainable parameters: $\theta_1, \theta_2$. The lift $\ell$ is the 'expanding window' lift of equation (6).

## 5   Conclusion

There is a strong corpus of theory motivating the use of the signature transform as a tool to understand streams of data. Meanwhile neural networks have enjoyed great empirical success. It is thus desirable to bring them together; in this paper we have described how this may be done in a general fashion, and have provided examples of how this principle may be used in a variety of domains.

There are two key contributions. First, we discuss stream-preserving neural networks, which are what allow for using signature transforms deeper within a network, rather than as just a feature transformation. Second, we discuss lifts, which is what allows for the use of multiple signature transforms. In this way we have significantly extended the use of the signature transform in machine learning: rather than limiting its usage to data preprocessing, we demonstrate how the signature transform, as a univeral nonlinearity, may be used as a pooling layer within a neural network.

**Acknowledgements**

PB was supported by the EPSRC grant EP/R513295/1. PK was supported by the EPSRC grant EP/L015811/1. PK, IPA, CS, TL were supported by the Alan Turing Institute under the EPSRC grant EP/N510129/1.

## Footnotes

[3]As $(d^{N+1} - 1)/(d - 1) = \sum_{k=0}^{N} d^k$ is the number of scalar values in a signature with $N$ terms.

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
