[Supplementary Material · DSig_appendix.pdf]

# A A brief overview of signatures

This appendix is split into three subsections. The first subsection discusses the definition and properties of the signature transform on path space, which is the mathematically natural way to approach things. The second subsection goes on to adapt the signature transform to the space of streams of data.

In the third subsection we discuss how to compute the signature transform. In particular we will see that whilst the signature transform of a path may look somewhat unfriendly to compute, it will (fortunately!) turn out that the signature transform may be efficiently computed in the special case that its input is piecewise linear, which is how a stream of data is interpreted.

## A.1 Signatures of paths

We begin with the definition of the signature. Note that this definition is written in a slightly different format to that of Definition 1.1, as the more traditional (if somewhat unfriendly-looking) notation of stochastic calculus is used; however the mathematical meaning is the same.

**Definition A.1** ([10]). Let $a, b \in \mathbb{R}$, and $X = (X^1, \ldots, X^d) \colon [a, b] \to \mathbb{R}^d$ be a continuous piecewise smooth path. The signature of $X$ is then defined as the collection of iterated integrals

$$\mathrm{Sig}(X) = \left( \int \cdots \int_{a < t_1 < \cdots < t_k < b} \mathrm{d}X_{t_1} \otimes \cdots \otimes \mathrm{d}X_{t_k} \right)_{k \geq 0}$$

$$= \left( \left( \int \cdots \int_{a < t_1 < \cdots < t_k < b} \mathrm{d}X_{t_1}^{i_1} \cdots \mathrm{d}X_{t_k}^{i_k} \right)_{1 \leq i_1, \ldots, i_k \leq d} \right)_{k \geq 0},$$

where $\otimes$ denotes the tensor product, $\mathrm{d}X_t = \frac{\mathrm{d}X_t}{\mathrm{d}t} \mathrm{d}t$, and the $k = 0$ term is taken to be $1 \in \mathbb{R}$.

The truncated signature of depth $N$ of $X$ is defined as

$$\mathrm{Sig}^N(X) = \left( \int \cdots \int_{a < t_1 < \cdots < t_k < b} \mathrm{d}X_{t_1} \otimes \cdots \otimes \mathrm{d}X_{t_k} \right)_{0 \leq k \leq N}.$$

The signature may in fact be defined much more generally, on paths of merely bounded variation, see [10, 11], but the above definition suffices for our purposes. This broader theory is also the reason for the notation $\mathrm{d}X_t$, which may be made sense of even when $X$ is not differentiable.

**Example A.2.** Suppose $X \colon [a, b] \to \mathbb{R}^d$ is the linear interpolation of two points $x, y \in \mathbb{R}^d$, so that $X_t = x + \frac{t-a}{b-a}(y - x)$. Then its signature is just the collection of powers of its total increment:

$$\mathrm{Sig}(X) = \left( 1, y - x, \frac{1}{2}(y - x)^{\otimes 2}, \frac{1}{6}(y - x)^{\otimes 3}, \ldots, \frac{1}{k!}(y - x)^{\otimes k}, \ldots \right).$$

Which is independent of $a, b$.

**Definition A.3.** Given a path $X \colon [a, b] \to \mathbb{R}^d$, we define the corresponding *time-augmented path* by $\widehat{X}_t = (t, X_t)$, which is a path in $\mathbb{R}^{d+1}$.

The signature exhibits four key properties that makes its use attractive when dealing with path-like data. First, a path is essentially defined by its signature. This means that essentially no information is lost when applying the signature transform.

**Proposition A.4** (Uniqueness of signature [43]). *Let $X \colon [a, b] \to \mathbb{R}^d$ be a continuous piecewise smooth path. Then $\mathrm{Sig}(\widehat{X})$ uniquely determines $X$ up to translation.*

Next, the terms of the signature decay in size factorially.

**Proposition A.5** (Factorial decay [10, Lemma 2.1.1])**.** *Let* $X \colon [a, b] \to \mathbb{R}^d$ *be a continuous piecewise smooth path. Then*

$$\left\| \int \cdots \int_{a < t_1 < \cdots < t_k < b} \mathrm{d}X_{t_1} \otimes \cdots \otimes \mathrm{d}X_{t_k} \right\| \leq \frac{C(X)^k}{k!},$$

*where $C(X)$ is a constant depending on $X$ and $\| \cdot \|$ is any tensor norm on $(\mathbb{R}^d)^{\otimes k}$.*

Third, functions of the path are approximately linear on the signature. In some sense the signature may be thought of as a 'universal nonlinearity' on paths.

**Proposition A.6** (Universal nonlinearity [44])**.** *Let $F$ be a real-valued continuous function on continuous piecewise smooth paths in $\mathbb{R}^d$ and let $\mathcal{K}$ be a compact set of such paths.*[4] *Furthermore assume that $X_0 = 0$ for all $X \in \mathcal{K}$. (To remove the transalation invariance.) Let $\varepsilon > 0$. Then there exists a linear functional $L$ such that for all $X \in \mathcal{K}$,*

$$\left| F(X) - L(\mathrm{Sig}(\widehat{X})) \right| < \varepsilon$$

Finally, the signature is invariant to time reparameterisations.

**Proposition A.7** (Invariance to time reparameterisations [10])**.** *Let $X \colon [0, 1] \to \mathbb{R}^d$ be a continuous piecewise smooth path. Let $\psi \colon [0, 1] \to [0, 1]$ be continuously differentiable, increasing, and surjective. Then $\mathrm{Sig}(X) = \mathrm{Sig}(X \circ \psi)$.*

Thus the signature encodes the *order* in which data arrives without caring precisely *when* it arrives; it is essentially factoring out the infinite-dimensional group of time reparameterisations. For example, consider the scenario of recording the movement of a pen as it draws a character on a piece of paper. Then the signature of the stream of data is invariant to the speed at which the character was drawn.

**Remark A.8.** There is an interesting interplay between Proposition A.6 and Proposition A.7. If one desires invariance to time reparameterisations, as in the example of a pen drawing a character, then computing the signature of just $X$ rather than $\widehat{X}$ will ensure by Proposition A.7 that this invariance is present. If one does not desire invariance to time reparameterisations, then using the time-augmented path $\widehat{X}$ is what ensures that parameterisation-dependent functions may still be learned. This essentially corresponds to the difference between $\widehat{X \circ \psi}$ and $\widehat{X} \circ \psi$.

## A.2 Signatures of streams of data

We interpret a stream of data as a discretisation of a path.

**Definition A.9.** The space of streams of data is defined as

$$\mathcal{S}(\mathbb{R}^d) = \{ \mathbf{x} = (x_1, \ldots, x_n) : x_i \in \mathbb{R}^d, n \in \mathbb{N} \}.$$

Given $\mathbf{x} = (x_1, \ldots, x_n) \in \mathcal{S}(\mathbb{R}^d)$, the integer $n$ is called the length of $\mathbf{x}$. Furthermore for $a, b \in \mathbb{R}$ such that $a < b$, fix

$$a = u_1 < u_2 < \cdots < u_{n-1} < u_n = b.$$

Let $X = (X^1, \ldots, X^d) \colon [a, b] \to \mathbb{R}^d$ be continuous such that $X_{u_i} = x_i$ for all $i$, and linear on the intervals in between. Then $X$ is called a linear interpolation of $\mathbf{x}$.

**Definition A.10.** Let $\mathbf{x} = (x_1, \ldots, x_n) \in \mathcal{S}(\mathbb{R}^d)$ be a stream of data. Let $X$ be a linear interpolation of $\mathbf{x}$. Then the signature of $\mathbf{x}$ is defined as

$$\mathrm{Sig}(\mathbf{x}) = \mathrm{Sig}(X)$$

and the (truncated) signature of order $N$ of $\mathbf{x}$ is defined as

$$\mathrm{Sig}^N(\mathbf{x}) = \mathrm{Sig}^N(X).$$

*A priori* this definition of the signature of a stream of data depends on the choice of linear interpolation. (That is, the speed at which one traverses the gap between the $x_i$.) However, it turns out that Definition A.10 is well-defined and independent of this choice, by Proposition A.7. See [10, Lemma 2.12].

**Remark A.11.** Let $\mathbf{x} = (x_1, \ldots, x_n) \in \mathcal{S}(\mathbb{R}^d)$ be a stream of data of length $n$ in $\mathbb{R}^d$. Then $\mathrm{Sig}^N(\mathbf{x})$ has

$$\sum_{k=0}^{N} d^k = \frac{d^{N+1} - 1}{d - 1}$$

components. In particular, the number of components does not depend on $n$; the truncated signature maps the infinite-dimensional space of streams of data $\mathcal{S}(\mathbb{R}^d)$ into a finite-dimensional space of dimension $(d^{N+1} - 1)/(d - 1)$. Thus the signature is an excellent way to tackle long streams of data, or streams of variable length, or streams for which certain data is missing.

## A.3   Computing the signature

Observe that the signature is defined as a sequence where the zeroth term is $1 \in (\mathbb{R}^d)^{\otimes 0} = \mathbb{R}$, the first term belongs to $\mathbb{R}^d$, the second term belongs to $\mathbb{R}^d \otimes \mathbb{R}^d$ (that is, the space of matrices of size $d \times d$), the third term belongs to $\mathbb{R}^d \otimes \mathbb{R}^d \otimes \mathbb{R}^d$ (that is, the space of tensors of shape $(d, d, d)$), and the $k$th term belongs to $(\mathbb{R}^d)^{\otimes k} = \mathbb{R}^d \otimes \cdots \otimes \mathbb{R}^d$, $k$ times (that is, the space of tensors of shape $(d, \ldots, d)$, $k$ times). With this description, the signature naturally takes values in the tensor algebra:

**Definition A.12.** The tensor algebra of $\mathbb{R}^d$ is defined as

$$T((\mathbb{R}^d)) = \prod_{k=0}^{\infty} (\mathbb{R}^d)^{\otimes k}.$$

The tensor product $\otimes$ is typically defined between two tensors, taking a tensor of shape $(a_1, \ldots, a_n)$ and a tensor of shape $(b_1, \ldots, b_m)$ to a tensor of shape $(a_1, \ldots, a_n, b_1, \ldots, b_m)$. For example, in the special case that these two tensors are of shapes $(a_1)$, $(b_1)$, so that they are vectors, then the tensor product is what is referred to as the *outer product*.

**Definition A.13.** When extended by bilinearity, the tensor product defines a multiplication on $T((\mathbb{R}^d))$. For $A = (A_0, A_1, \ldots) \in T((\mathbb{R}^d))$ and $B = (B_0, B_1, \ldots) \in T((\mathbb{R}^d))$, then $A \otimes B \in T((\mathbb{R}^d))$ can be seen to be

$$A \otimes B = \left( \sum_{j=0}^{k} A_j \otimes B_{k-j} \right)_{k \geq 0}.$$

A fundamental insight of Chen is that concatenation of paths corresponds to tensor multiplication of their signatures. The following relation is known as *Chen's identity*.

**Proposition A.14** (Chen's identity, [10, Theorem 2.12]). *Let* $X \colon [a, b] \to \mathbb{R}^d$ *and* $Y \colon [a, b] \to \mathbb{R}^d$ *be two continuous piecewise smooth paths such that* $X_b = Y_a$. *Define their concatenation* $X * Y$ *as*

$$(X * Y)_t = \begin{cases} X_{2t-a} & \text{for} \quad a \leq t < \dfrac{a+b}{2}, \\[2mm] Y_{2t-b} & \text{for} \quad \dfrac{a+b}{2} \leq t \leq b. \end{cases}$$

*Then*

$$\mathrm{Sig}(X * Y) = \mathrm{Sig}(X) \otimes \mathrm{Sig}(Y).$$

Equipped with Chen's identity, the signature of a stream is straightforward to compute explicitly.

**Proposition A.15.** *Let* $\mathbf{x} = (x_1, \ldots, x_n) \in \mathcal{S}(\mathbb{R}^d)$ *be a stream of data. Then,*

$$\mathrm{Sig}(\mathbf{x}) = \exp(x_2 - x_1) \otimes \exp(x_3 - x_2) \otimes \cdots \otimes \exp(x_n - x_{n-1}),$$

*where*

$$\exp(x) = \left( \frac{x^{\otimes k}}{k!} \right)_{k \geq 0} \in T((\mathbb{R}^d)).$$

*Proof.* It is easy to check that if $\mathbf{x} = (x_1, x_2) \in \mathcal{S}(\mathbb{R}^d)$ is a stream of data of length 2 then the signature of $\mathbf{x}$ is given by $\exp(x_2 - x_1)$, as in Example A.2. So given a stream of data $\mathbf{x} = (x_1, \ldots, x_n) \in \mathcal{S}(\mathbb{R}^d)$ of length $n \geq 2$, iteratively applying Chen's identity yields the result. $\qquad\square$

**Remark A.16.** Chen's identity implies that computing the signature of an incoming stream of data is efficient. Indeed, suppose one has obtained a stream of data $\mathbf{x} \in \mathcal{S}(\mathbb{R}^d)$ and computed its signature. Suppose that after some time more data has arrived, $\mathbf{y} \in \mathcal{S}(\mathbb{R}^d)$. In order to compute the signature of the whole signal one only needs to compute the signature of the new piece of information, and tensor product it with the already-computed signature.

**Remark A.17.** Computing signatures in the manner described here involves only normal tensor operations, so it may be backpropagated through in the usual way. Recall that signatures are fundamentally defined on path space; backpropagating corresponds to determining the perturbation of the signature when perturbing its input with white noise. However one of the insights of *rough path theory* [10] is that a path needs more than just its pointwise values to be fully determined. The most common example of this arises in stochastic calculus, where one has to make a choice between Itô and Stratonovich integration. Until such a choice is made, one cannot define a notion of integrals of the path. In general, for sufficiently rough paths, one has to *define* what the integrals of a path are: essentially the path is defined by its signature, rather than the other way around. In such a framework it is not clear what the correct notion of perturbations of path space are, and this remains a direction for future work.

# B   Implementation Details

All experimental models were trained using the Adam [45] optimiser as implemented by PyTorch [46], which was the framework used to implement the models. Signature calculations were performed with the `iisignature` package [47] (as the Signatory project [22] mentioned elsewhere in this paper had not yet been developed). All activation functions were taken to be the ReLU. Computations were performed on two computers. One was equipped with two Tesla K40m GPUs. The second was equipped with two GeForce RTX 2080 Ti GPUs and two Quadro GP100 GPUs.

In each of the following sections, the notation is the same as the notation used in the corresponding section of the main document.

## B.1   A generative model for a stochastic process

The training dataset was given by 1024 realisations of an Ornstein–Uhlenbeck process, and the test set was of the same size, each sampled at 100 points of $[0, 1]$. No minibatching was used. The model was trained for 500 epochs.

The layer $\Phi^{\theta_1}$ operated pointwise on the stream of time-augmented Brownian motion $\mathbf{B} = (t_i, B_{t_i})_i \in \mathcal{S}(\mathbb{R}^2)$, and was taken to be a neural network with 2 output neurons and 2 hidden layers of 8 neurons. Furthermore it kept the original stream; thus

$$\Phi^{\theta_1}(\mathbf{B}) = (t_i, B_{t_i}, \phi_1^{\theta_1}(t_i, B_{t_i}), \phi_2^{\theta_1}(t_i, B_{t_i})) \in \mathcal{S}(\mathbb{R}^4)$$

for some learned $\phi_1^{\theta_1}, \phi_2^{\theta_1}$. The lift was the 'expanding window' described in equation (6). The signature in the generator was truncated at $N = 3$ (giving 84 scalar nonconstant terms) The layer $f^{\theta_2}$ operated pointwise on the stream of signatures, and was a simple linear map down to a scalar value (the value of the generated process at that time step). The signature in the discriminator was truncated at $M = 4$.

Some hyperparameter searching was necessary to obtain good results. The search was not done according to any formal scheme. It seemed that if $\Phi^{\theta_1}$ was sufficiently simple and not did not keep the original stream then the training would easily get trapped in a bad local minima, and the generated process would be visually distinct from the Ornstein–Uhlenbeck process.

## B.2   Supervised learning with fractional Brownian motion

The training set featured 600 samples whilst the test set featured 100 samples, each of an instance of fractional Brownian motion sampled at 300 time steps of $[0, 1]$, with Hurst parameters in the range

$[0.2, 0.8]$. These were split up into batches of 128 samples, so the last batch is slightly smaller than the others, and every model trained for 100 epochs. The loss function was taken to be mean squared error (MSE).

There was no hyperparameter searching except to require that all models should have approximately the same number of parameters; in all cases the results represent a model whose hyperparameters have not been fine-tuned to the task at hand.

All models used a sigmoid as a final nonlinearity, so as to map in to $(0, 1)$.

The differing sizes of layers between models (whilst keeping roughly the same overall parameter count) is usually because of the varying size of the input to the model. Some models take all of the raw data, some models use signatures, and some models take expanding or sliding windows of the data in a manner akin to equations (6) and (8).

The Feedforward model was a simple neural network with 3 hidden layers of 16 neurons each.

The Neural-Sig model – which is essentially the same model as the Feedforward model, except that the data has the signature applied as feature transformation first – featured hidden layers of sizes 64, 64, 32, 32, 16, 16 respectively.

The RNN model is two recurrent neural networks, the first comprised of dense layers of sizes 64, 64, 32 and output size 6, and the second comprised of dense layers of size 32, 32, 32, and output size 5. The first network sweeps across the input data relatively slowly, with a stride of 2, whilst the second network sweeps across the result of the first network more quickly, with a stride of 4. In this way it may capture information from the input data at multiple timescales; part of the challenge of fractional Brownian motion is the existence of long-range dependencies [48].

The LSTM and GRU models both featured two recurrent layers each of size 32, and swept across the raw data with a stride of 1.

DeepSigNet featured a single Neural-Lift-Signature block, where the neural component was given by a single convolutional layer with 3 channels and kernel size 3, the lift was the trivial lift of equation (9), and the signature was truncated as $N = 3$. The neural component also preserved the original time-augmented stream of data, so that in some sense the neural component has 3 extra channels corresponding to time and value. On top of this a feedforward neural network with 5 hidden layers of size 32 was placed. Thus this model is very similar to the Neural-Sig model, except that a small learnable transformation was allowed before the signature. The difference in their performance highlights the value of learning a transformation before using the signature. (Without which the Neural-Sig model is merely outperformed by some non-signature based models.)

DeeperSigNet featured three Neural-Lift-Signature blocks. The neural component of the first block was a small feedforward network with 2 hidden layers of size 16 and an output layer of size 3, swept across the length of the stream; its kernel size (how many time-value pairs of the stream it saw at once) was 4. The original time-augmented stream of data was also preserved by the neural component. The neural components of the other two blocks were recurrent neural networks, featuring 2 hidden layers of 16 neurons each. The lifts were in every case expanding windows as in equation (6). On top of this another recurrent neural network was placed, and the value of its final hidden state used as the output. This final network used 2 hidden layers of 16 neurons each.

### B.3   Non-Markovian deep reinforcement learning

We used the implementation of the Mountain Car problem implemented by the OpenAI Gym [41], modified to return only the car's position. Each episode was run for 300 steps, and each model was given 2000 episodes in which to learn. The reward function was given by the car's position, in the range $(-1.2, 0.6)$, with a bonus $+1$ on reaching the goal. At each step the car could drive its engine left, right, or not use it at all. This problem was chosen for its ease of implementation.

The sizes of the models were chosen to ensure that they both had roughly the same number of scalar parameters. Within this specification, there was a small amount of hyperparameter searching. This was done in an *ad hoc* manner, for both models, varying the number of layers and the numbers of neurons in each layer, around the values that were eventually used. The eventual values chosen for the deep signature model were selected as the ones giving the best results for the deep signature

model. The eventual values for the RNN were selected to give roughly the same parameter count as the deep signature model, as no RNN model achieved any appreciable success.

The deep signature model was as described in Section 4.3, with the first network $\Phi^{\theta_1}$ applying a learned linear transformation with output dimension 2. Furthermore it kept the original time-augmented stream, so that

$$y_i = \Phi^{\theta_1}(x_i) = (t_i, x_i, \phi_1^{\theta_1}(t_i, x_i), \phi_2^{\theta_1}(t_i, x_i)) \in \mathcal{S}(\mathbb{R}^4),$$

where $\phi_1^{\theta_1}$ and $\phi_2^{\theta_2}$ are learned linear functions. The signature was truncated at $N = 3$. The second network $f^{\theta_2}$ was comprised of a single hidden layer of 64 neurons, followed by an output layer of 3 neurons, corresponding to the three possible actions. The action with the greatest value was the action selected. This model had a total of 5769 scalar parameters.

The RNN model featured 3 recurrent layers each of size 32, followed by an output layer of 3 neurons, corresponding to the three possible actions. The action with the greatest value was the action selected. This model had a total of 5475 scalar parameters.

The reinforcement learning technique used was Deep Q Learning [42, 49], to effectively transform the task into a supervised learning problem, with each of the specified models performing function approximation on $Q$. Actions were chosen in an $\varepsilon$-greedy manner, with $\varepsilon = 0.2$. The discount factor was given by $\gamma = 0.99$.

The deep signature model achieved success, and would learn to consistently solve the problem at around 1500 episodes. The RNN failed to achieved success within 2000 episodes on any test run. 3 test runs were performed.

## Footnotes

[4]Of course the definition of both continuity and compactness depend on the topology of the set of paths. See [21] for details.