[Reviews · NeurIPS 2019]

Reviewer 1



Summary ======= The authors propose to use a already known method as a pooling function for time series. The idea is to leverage an integral transform, the *path signature*, to map a discretized curve on a real valued sequence. Truncation leads to a vectorized representation which is then used in practice. The proposed transformation is differentiable and thus can be integrated into models trained via backpropagation in an end-to-end manner. As the transformation consumes the one dimension, i.e. the time, stacking it requires to reintroduce a time series like structure in the output. One way of doing so is discussed by the authors. Finally, the application is evaluated on synthetic datasets in the context of (1) learning a generative model, (2) supervised learning of a system hyper-parameter, (3) reinforcement learning. Originality =========== Applying functional transforms as pooling is not a new idea. However, using path signatures as introduced in this work may be beneficial for certain applications. Quality ======= The overall notation is good and easy to follow. Maybe a little more effort could have been spent on introducing the path signature, e.g., the notion of a tensor product is ambiguous and thus should be explicitly defined. There are two major issues which have to be specifically addressed: (1) The lack of motivation. There are many ways of computing signatures of curves, indeed every spectral representation, for example. I do not the *selling point* for the path signature. Are there striking benefits over other transformations? Correlated to this, the related work section is stunningly short. Are there indeed no other approaches using some sort of functional transformation as intermediate differentiable function within the deep learning framework? (2) There is no conclusion! Given that the motivation is somehow assumed to be granted I would at least expect a conclusion summarizing the contributions and insights. Clarity ======= The overall clarity is good. The figures are well executed and contribute to the readers understanding. Significance ============ The authors do not outline and discuss the theoretical benefits of the path signature (although emphasizing the rich theoretical body behind it). In this context the contribution boils down to applying an already known functional transform as intermediate layer in a deep model and showing that it can outperform on not-benchmark synthetic datasets. From my perspective this sums up to a rather low contribution/significance.

Reviewer 2



The paper presents an interesting approach that allows signatures to be used as a layer anywhere within a neural network. The development of the method is well justified by both logical and theoretical argument and is technically sound. Simple as is, the proposed framework has revised the traditional feature transformation based Signature usage, and enriched the body of NN research. Moreover, the discussion about inverting signatures and generative model provides further insights.

Reviewer 3



Originality: This paper proposes a method to use the signature transform as a layer of neural networks. If this is the first work which successfully integrates the signature transform into deep learning, the novelty is high. The previous studies and the motivation of the work are well introduced. Quality: The way the signature transform is integrated into neural networks technically sounds. This seems to be an early work and the experiments are not very extensive. It is basically like "this works". Therefore, it is not very clear if the proposed method is meaningfully better in the real world applications. It would be great to have better explanations on what types of problems can be solved better with the proposed method than existing ones, and it is confirmed by experiments (i.e., more things like Figure 6 and Table 1). Clarity: The writing looks good to me. The experimental section is too brief and it could be improved. The paper needs to have conclusions. It was not clear how the sigmas in both equations in Section 6 relate each other. Significance: I think this is important work. If the signature transformation can be used as a layer and it has unique and important characteristics which are not trivial to represent by other means, the significance of the work is large.

[Author Response · NeurIPS 2019]

We thank the reviewers R1, R3 and R4 for their time and for their feedback.

**Motivation:** R1 expresses concern as to the 'selling point' of the signature transform, over other transformations; R4
expresses a similar concern about scenarios in which one would use the signature transform. We propose to add the
following paragraphs to Section 1.

*In multimodal data it can happen that the different channels represent linked information, and that the order of the*
*events in the different channels is the feature of interest. For example, regularly seeing the sequence: phone call, trade,*
*price movement in the stream of office data monitoring a trader might lead one to suspect insider trading.*

*Such occurrences are straightforward to detect with a regression on a few terms in the signature. This approach is*
*non-parametric and makes no attempt to model the original signal. Modelling this signal using Fourier series or*
*wavelets would be much more expensive: linearity of these transforms imply that each coordinate must be resolved*
*accurately enough to see the order of events.*

*The fundamental difference between the signature transform and classical signal transforms such as Fourier trans-*
*forms and wavelets is that the latter are used to model a parametrised version of a curve as a linear combination in*
*a functional basis. The signature does not try to model or parameterise the curve itself, but instead provides a basis*
*for functions on the space of curves. From a signal processing perspective, the signature can be thought of as a filter*
*which is invariant to resampling of the input signal.*

Certainly other transformations may be worth embedding within neural networks; it is the purpose of our paper to
demonstrate how this aim may be accomplished in this particular case. A full comparison of the different transforms
that may be selected would be the domain of another paper entirely. Furthermore an understanding on how to embed
the signature transform within neural networks, such as our paper, would be a prerequisite for such an investigation.

**Conclusion:** Both R1 and R4 requested a conclusion. We propose to add the following to the end of the paper.

*There is a strong corpus of theory motivating the use of the signature transform as a tool to understand streams of*
*data. Meanwhile neural networks have enjoyed great empirical success. It is thus desirable to bring them together;*
*in this paper we have laid out the theory describing how this may be done in a general fashion, and have provided*
*examples of how this principle may be used in a variety of domains.*

*There are two key contributions. First, we discuss stream-preserving neural networks, which is what allows for using*
*signature transforms deeper within a network, rather than as just a feature transformation. Second, we discuss lifts,*
*which is what allows for the use of multiple signature transforms. In this way we have significantly extended the use*
*of the signature transform in machine learning: rather than limiting its usage to data preprocessing, we demonstrate*
*how the signature transform, as a univeral nonlinearity, may be used as a general layer within a neural network.*

**Related Work:** R1 notes that our discussion of related work is essentially confined to the use of the signature trans-
form, as opposed to other functional transformations. We agree that this is lacking, and propose to add references to
the use of wavelets and Fourier transforms with neural networks to Section 2.

R4 remarks that "If this is the first work which successfully integrates the signature transform into deep learning, the
novelty is high". To the best of the authors' knowledge this is indeed the case. For completeness it is worth noting the
existence of the unpublished paper *Learning stochastic differential equations using RNN with log signature features*
by Liao, Ni, Lyons, and Yang, which was developed concurrently with our work. It uses a related transformation (the
log-signature) in a similar differentiable manner, but lacks the generality with which we combine signatures and neural
networks; they focus instead on a particular application.

**Experiments:** R3 asks to restructure the presentation of the evaluation part. We are not certain precisely where their
concerns lie, but will keep their concern in mind when incorporating the other changes.

R4 comments that it would be nice to have more extensive experiments. We agree, but were space-limited, and
decided to focus on demonstrating the breadth of applications - generative, supervised, reinforcement - rather than
just producing the usual paper demonstrating good results on just supervised learning problems. Perhaps not directly
applicable, but we would like to note that the related work cited within Sections 1 and 2 already demonstrate excellent
results using the signature transform in multiple types of supervised learning problems, albeit whilst using the signature
transform only in the feature transformation-based manner.

**Use of $\sigma$:** R4 comments that the relationship between the $\sigma$ in Section 6 is unclear. We believe they are referring to the
derivation of the equation preceding line 244 from the equation preceding line 242. We propose to fix this by adding
a brief reference to Chen's identity, as described in Appendix A, from which this derivation follows immediately.

We thank R3 for their positive support. We hope that the changes proposed above satisfy the improvements requested
by R1 and R4.

[Meta-Review · NeurIPS 2019]

This paper proposes a method to incorporate signature transform as a layer of a neural network. The proposed scheme is empirically backed up in different scenarios (supervised learning, learning generative model, and RL). Majority of reviewers find the contributions of this submission significant. So I recommend accept, but I ask authors to clarify, in the final version, the novelty concerns that R1 has raised.